# A global dataset of high-resolution CO<sub>2</sub> enhancements derived from OCO-3 measurements

Pingyu Fan<sup>1,6</sup>, Jiangong Liu<sup>2</sup>, Yuan Xu<sup>3</sup>, Bo Huang<sup>4</sup>, Chris Webster<sup>5</sup>, and Yulun Zhou<sup>1,6,7</sup>

<sup>1</sup>Department of Urban Planning and Design, The University of Hong Kong, Hong Kong, China

<sup>2</sup>Department of Earth and Environmental Engineering, Columbia University, New York, United States

<sup>3</sup>Department of Geography and Resources Management, The Chinese University of Hong Kong, Hong Kong, China

<sup>4</sup>Department of Geography, The University of Hong Kong, Hong Kong, China

<sup>5</sup>Urban Systems Institute, The University of Hong Kong, Hong Kong, China

<sup>6</sup>The University of Hong Kong Shenzhen Institute of Research and Innovation, Shenzhen, China

<sup>7</sup>Musketeers Institute of Data Science, The University of Hong Kong, Hong Kong, China

Correspondence: Yulun Zhou (yulunzhou@hku.hk)

Abstract. We present a novel global dataset of  $CO_2$  enhancements ( $\Delta XCO_2$ ) derived by fusing NASA's OCO-3 satellite and NOAA ground-based observations.  $CO_2$  enhancements quantify the spatially resolved excess in atmospheric  $CO_2$  concentrations arising from anthropogenic emissions, biospheric  $CO_2$  exchanges, and atmospheric  $CO_2$  transport. Leveraging decades of monthly  $CO_2$  measurements from eight remote stations strictly selected from NOAA ESRL network, such as the Mauna Loa

- station, we address the critical challenge of isolating localized CO<sub>2</sub> signals from background concentrations by developing a latitude-dependent global CO<sub>2</sub> baseline model that effectively captures spatial and seasonal variability in background CO<sub>2</sub>. The developed baseline model demonstrates near-perfect hemispheric predictive accuracy (Northern:  $R^2$ =0.988, RMSE=1.78 ppm; Southern:  $R^2$ =0.995, RMSE=1.09 ppm). Spatially explicit  $\Delta XCO_2$  is then estimated by removing the column-corrected background CO<sub>2</sub> from co-located OCO-3 observations. Validations of the estimated  $\Delta XCO_2$  against tropospheric NO<sub>2</sub> ( $R^2$ =0.896)
- and prior in-situ urban CO<sub>2</sub> measurements, along with the dataset's high spatiotemporal resolution ( $\sim 3 \text{ km}^2$ ), demonstrates its potential for tracking anthropogenic and biospheric CO<sub>2</sub> dynamics. Global  $\Delta$ XCO<sub>2</sub> maps reveal mean CO<sub>2</sub> enhancements of 0.58 ± 1.81 ppm, with urban areas exhibiting 1.5-fold higher enhancements (1.43 ± 2.04 ppm). North Hemisphere land areas exhibits an approximately 81% higher  $\Delta$ XCO<sub>2</sub> average (0.67 ± 1.98 ppm) compared to the South Hemisphere (0.37 ± 1.32 ppm), with urban enhancements amplifying this hemispheric contrast up to 95%. Comprising 54 million observations
- across more than 200 countries, this open-access dataset provides an alternative metric for monitoring complex atmospheric CO<sub>2</sub> variability and actionable insights for regional climate policies, available at https://doi.org/10.5281/zenodo.15209825.

#### 1 Introduction

Mapping global CO<sub>2</sub> enhancements with fine spatio-temporal resolution is essential for tracking anthropogenic and natural CO<sub>2</sub> sources, validating emission inventories and climate models, and assessing localized climate impacts, including effects on agricultural productivity. The CO<sub>2</sub> enhancement ( $\Delta$ XCO<sub>2</sub>) quantifies the combined influence of anthropogenic carbon

emissions, the net CO<sub>2</sub> exchange between vegetation and the atmosphere, and large-scale atmospheric transport processes

(Kiel et al., 2021; Mitchell et al., 2018; Reuter et al., 2019; Lei et al., 2022). A positive  $\Delta XCO_2$  signifies the net positive impacts of CO<sub>2</sub> sources and sinks with atmospheric transport processes on accumulation of atmospheric CO<sub>2</sub> concentration (Park et al., 2021; Kiel et al., 2021; Mitchell et al., 2018).  $\Delta XCO_2$  can become negative when enhanced terrestrial CO<sub>2</sub> uptake surpassing the amount emitted and atmospheric conditions such as prevailing wind result in a net decline in atmospheric CO<sub>2</sub> compared to long-term baseline. Accurate global  $\Delta XCO_2$  measurements are critical for understanding the current net

- contributions of local human and plant activities on atmospheric  $CO_2$  increases. These fundamental measurements underpin the development of targeted strategies for future emission cuts and climate mitigation. Nevertheless, a comprehensive global dataset of  $CO_2$  enhancements remains absent.
- Carbon satellites have been reported as objective, independent data sources for monitoring spatiotemporal disparities of atmospheric  $CO_2$  conditions (Pan et al., 2021; Nisbet and Weiss, 2010; Schwandner et al., 2017), by providing the top-down observations of column-averaged dry-air mole fraction of  $CO_2$  (XCO<sub>2</sub>; ppm). Satellite-derived  $CO_2$  observations enhance data disclosure, transparency, and data equity especially in underdeveloped countries where data infrastructure is lacking and the accounting capacity of environmental departments is often weak. Moreover, the global coverage and high spatiotemporal
- resolution of satellite instruments are key advantages that support the characterization of large-scale fine-grained atmospheric  $CO_2$  levels. These capabilities, first, enable effective comparisons among multiple sites, cities, and countries; second, help refine the general patterns underlying local  $CO_2$  variability, and ultimately, aid  $CO_2$  mitigation efficacy by facilitating local governments making dynamic and targeted decisions.
- Mapping  $\Delta XCO_2$  based on satellite remote-sensing data remains challenging, although CO<sub>2</sub> satellites have been popular for 40 global atmospheric observations (Streets et al., 2013). The century-long persistence of CO<sub>2</sub> causes the immensely strong signal and notable spatial variability of background CO<sub>2</sub> concentration in the atmosphere, even in desert-like places (Hakkarainen et al., 2019). This accumulation masks the true CO<sub>2</sub> signals from local human and natural processes. Localized atmospheric CO<sub>2</sub> fluctuations are around two orders of magnitude smaller than the background CO<sub>2</sub> concentration (Canadell et al., 2023; Reuter et al., 2019). This substantial difference complicates the global differentiation of enhanced CO<sub>2</sub> signals from the accu-
- mulative trend of background CO<sub>2</sub>. Therefore, a key step in isolating localized  $\Delta XCO_2$  is deducting accurate fine-resolution background CO<sub>2</sub> concentration from the satellite CO<sub>2</sub> observations.

Satellite-driven  $\Delta XCO_2$  measurements are typically single site-specific, individual city-specific, or estimated from multiplesite gradients, rather than offering spatially-continuous coverage. Previous efforts to correct background CO<sub>2</sub> concentration have struggled with scale and seasonal sensitivity (Lindenmaier et al., 2014; Verhulst et al., 2017; Zeng et al., 2021; Miller et al.,

- 2020; Che et al., 2024; Kort et al., 2012; Schneising et al., 2013). These issues constrain the effectiveness in delivering largescale spatial analysis for fine-resolution  $\Delta XCO_2$  data. Site-specific  $\Delta XCO_2$  estimates are derived from the difference between satellite observation of atmospheric CO<sub>2</sub> and CO<sub>2</sub> records at ground-truth stations with minimal human and plant interference. These ground-truth stations include remote in-site stations from the Total Carbon Column Observing Network (Kiel et al., 2021) and mountaintop CO<sub>2</sub> observation sites in Salt Lake County, North America (Mitchell et al., 2018). City-specific  $\Delta XCO_2$
- estimates rely on the deviations of satellite observations in urban centers from the daily median (Hakkarainen et al., 2016; Park et al., 2021) or monthly median (Labzovskii et al., 2019) remote-sensing  $CO_2$  observations in rural areas assuming that rural

atmospheric CO<sub>2</sub> concentration should be considerably lower than that in urban cores (Wu et al., 2020; Reuter et al., 2019; Ye et al., 2020). The former site-specific approach cannot be effectively generalized for regional or global applications due to significant spatial heterogeneity in atmospheric conditions. The latter city-specific approach is sensitive to the seasonal timing of satellite overpasses, since seasonal biospheric-atmospheric CO<sub>2</sub> fluxes obscure the differentiation of urban-derived  $\Delta$ XCO<sub>2</sub> signals from background CO<sub>2</sub> concentration in the rural areas. This limited generalizability can be in large part improved by

- signals from background CO<sub>2</sub> concentration in the rural areas. This limited generalizability can be in large part improved by leveraging the cooperative air sampling network of the National Oceanic and Atmospheric Administration (NOAA), which provide a range of remote marine stations for the developments of latitudinal references for background CO<sub>2</sub> concentration (Masarie and Tans, 1995; Tans et al., 1989). However, the continental-to-global spatial scales along multiple-site gradients of atmospheric background CO<sub>2</sub> (Mitchell et al., 2018) remain too coarse for precisely tracking background CO<sub>2</sub> dynamics
- in cities. This underscores the need for more spatially-explicit measurement techniques to better characterize subtle  $\Delta XCO_2$  variations within cities that contribute significant shares to global emissions (Duren and Miller, 2012).

To present both globally comprehensive and locally representative measurements of  $\Delta XCO_2$ , we leverage a NASA's new Orbiting Carbon Observatory 3 (OCO-3) satellite that offers the state-of-the-art highest resolution observations ( $\approx 3 \text{ km}^2$ )

covering the period from August 2019 to November 2023, and further remove atmospheric background  $CO_2$  corrected from a novel global  $CO_2$  baseline model based on ground-sourced  $CO_2$  data from the Global Monitoring Laboratory (NOAA ESRL network). The satellite-derived dataset of global  $CO_2$  enhancements enables objective, timely and spatially-explicit diagnosis of net impacts of  $CO_2$  sources, sinks, and transport on atmospheric  $CO_2$  increases, contributing to sub-city scale decision making on global net-zero strategies and climate actions.

#### 75 2 Methods

This work develops a novel dataset of global CO<sub>2</sub> enhancements from 2019 to 2023 by integrating satellite-derived and groundsourced CO<sub>2</sub> observations. Fig. 1 demonstrates the dataset involved and the main workflows. The following three subsections elaborate on carbon satellite XCO<sub>2</sub> product (XCO<sub>2</sub>), the global CO<sub>2</sub> baseline estimation (CO<sub>2b</sub>), and the global CO<sub>2</sub> enhancements ( $\Delta$ XCO<sub>2</sub>), respectively.

# 80 2.1 Satellite-retrieved XCO<sub>2</sub> Observations

Satellite-derived XCO<sub>2</sub> observations (column-averaged dry air mole fraction of CO<sub>2</sub>; ppm) are from the Orbiting Carbon Observatory 3 (OCO-3). This NASA satellite, launched in 2019, collects the global magnitude and distribution of atmospheric CO<sub>2</sub> concentrations with the highest spatial-temporal resolution to date, allowing it to track XCO<sub>2</sub> variations with a grid resolution of  $2.2 \times 1.6$  km<sup>2</sup>. Studies have proven that the OCO-3 is capable of detecting localized emission sources by giving

85 diurnal and geographically diverse XCO<sub>2</sub> observations (Kiel et al., 2021; Schwandner et al., 2017) and are less vulnerable to the impacts of small-scale atmospheric processes on the accuracy of local emission accounting (McKain et al., 2012). We use OCO-3 Level 2 bias-corrected XCO<sub>2</sub>, version 10.4r data (OCO3\_L2\_Lite\_FP), which is publicly available through the NASA Goddard Earth Science Data and Information Services Center (GES DISC) (http://disc.sci.gsfc.nasa.gov/). We filter the OCO-

#### **Input Datasets**