# Peer review of "A global dataset of high-resolution CO2 enhancements derived from OCO-3 measurements"

_Earth System Science Data, 2025_

## Referee Comment (RC1)

**Review: A global dataset of high-resolution $CO_2$ enhancements derived from OCO-3 measurements**

by Pingyu Fan, et al.
September 30, 2025

**1 Review Summary**

In this work, the authors have used NOAA measurements from 8 surface stations to create a background globally-gridded carbon-dioxide ($XCO_{2b}$) field using a mathematical model. Differences against total column carbon-dioxide ($XCO_2$) observations made by NASA's Orbiting Carbon Observatory -3 (OCO-3) have been used to create an $XCO_2$ enhancement: $\Delta XCO_2 = XCO_2 - XCO_{2b}$. For an example case over Los Angeles, California, U.S.A., $\Delta XCO_2$ is shown to be well correlated against TROPOMI's $NO_2$ product, which is co-emitted with fossil fuel burning. Spatial maps of the gridded data product suggest enhancements over expected regions with higher population densities, i.e., urban areas.

The paper is well written in clear English, and the organization of the material is logical. The subject matter is appropriate for Copernicus's Earth System Science Data. However, due to some serious technical deficiencies (see major comments below) I believe that the article should not be published in its current form. My overall recommendation is to reject based on the comments given below. I would be willing to re-review a second iteration of the article if some of the major concerns could be addressed.

**2 Major Comments**

- In Sec.2.2.2 the authors discuss the calculation of the $CO_2$ baseline using fits to the 8 NOAA ground-based stations. It is unclear to me if the baseline product contains a longitudinally-dependent term or if the $CO_2$ is fixed for all longitudes. The mathematical model described in Eqs. 1-3 only depend on latitude (l) and time (m). While to first order this simplistic model can provide a reasonable guess as to the $CO_2$ at a given time and place, and could therefore provide a useful prior for a Bayesian inversion algorithm, the model is deficient with respect to examining small deviations from the background as would be found over urban areas (a few parts per million out of approximately 420 ppm). I believe that this is a fundamental flaw of the paper, and that the problem cannot be overcome without introducing the complication of involving atmospheric transport into the problem.

- Sec.2.2.2. This model sounds very similar to that used in (Lindqvist, 2015) to model and compare the seasonal cycle of $XCO_2$ between the GOSAT satellite and ground-based TCCON observations. However, in the current work, it seems that they use a single linear growth rate, which mean that they do not include interannual variability. They assume the $CO_2$ background can be described with just a single sinusoid plus a linear trend. From (Lindqvist, 2015) it was shown that the seasonal cycle is skewed/asymmetric (strong spring drawdown, slower buildup in fall). Also the large $R^2$ seem a bit misleading, as it is probably due to the large dynamic range. It might be better to check if the residuals show patterns/biases.

- While the $\Delta XCO_2$ data product in its current form is of some interest, it is not suitable for making detailed claims about $CO_2$ sources and sinks, and therefore not appropriate for informing policy. Determining sources and sinks of $CO_2$ at large scales (meso to global), which includes contributions from biogenic, fires, and anthropogenic sources, requires the use of flux inversion models which take into account long-range atmospheric transport, e.g., (Houweling, 2015; Byrne, 2023). It's true that estimates of fossil fuel emissions are being made at urban scales using various techniques, e.g., (Ye, 2020; Wu, 2022; Roten, 2023), but all of these techniques require at least meteorological conditions, e.g., wind speed and direction, usually taken from reanalysis weather models.

- The work could be made more general and useful if the authors made available the background $XCO_2$ dataset that they have created, which in theory could then be differenced against sensors other than OCO-3, e.g., OCO-

2 and/or GOSAT-1 and GOSAT-2. Their background product is calibrated to OCO-3, but given that OCO-2 and OCO-3 agree extremely well (Taylor, 2023), the methodology should work fine with OCO-2 (and hopefully GOSAT, GOSAT-2, etc.) equally well.

- The authors should compare and contrast their new background $XCO_2$ product against the few others that are publicly available. Other $CO_2$ backgrounds that the reviewer is aware of are the TCCON prior, which is a full profile (Laughner, 2023, 2024), and the NOAA background product (Rastogi, 2021), which is available only over North America. The authors should mention these products, and discuss how they differ from their own background, and why their new product is better for the purpose of creating $\Delta XCO_2$ enhancements.

- The NOAA ground-based observations are known to be highly precise and accurate. However, they fundamentally measure a different thing ($CO_2$ at the surface) as compared to the full column measurement from the satellite (Pandey, 2024). There is some concern that creating the $\Delta XCO_2$ product based on these 2 entities could mis-represent real $CO_2$ enhancements. The conversion of the ground-based NOAA $CO_2$ values to total column is discussed in Sec.2.3. This section is very brief and simply states that the ground-based NOAA observations were linearly scaled to collocated OCO-3 observations using fitting coefficients. I think it is important to provide a more in-depth analysis here since this is a critical step in the calculation. For example, what exactly were the spatiotemporal collocation criteria (distance and time between observations), and why not provide a table of the coefficients? Presumably these coefficients would change with OCO-3 v11 $XCO_2$ product, and as mentioned previously, could not this methodology be also applied to OCO-2 v11.2? OCO-2 does not provide the Snapshot Area Mapping mode observations like OCO-3, but it does have some Target observations over urban areas, especially in more recent years, as well as the nominal nadir and glint overpasses that sometimes intersect cities.

- There is no mention of an averaging kernel (AK) correction in the current work. The AK correction accounts for differences in the vertical sensitivity between sets of observations made by different instruments or models. I think an AK correction of the full column profiles calculated from the NOAA data are needed. Note that the corrections are typically small, on the order of 0.1 to 0.2 ppm, but occasionally they may be closer to 0.5 to 1.0 ppm. Information about AK corrections can be found in (O'Dell, 2018; Nguyen, 2020), among other places.

- Fig 7. The choice of cities that are reported are odd. I would recommend changing to some more globally well known cities such as: [https://en.wikipedia.org/wiki/List_of_largest_urban_areas_by_continent](https://en.wikipedia.org/wiki/List_of_largest_urban_areas_by_continent) and provide estimates of the city population for reference.

- Sec. 4.2: It would be helpful for the authors to compare/contrast their $XCO_2$ enhancements over urban areas with an emissions inventory like ODIAC: [https://www.odiac.org/index.html](https://www.odiac.org/index.html). I'm sure that there will be some reasonable correlation, but I would expect the signals from OCO to look spatially "washed out" compared to the high spatial resolution of ODIAC. This is due to emissions of $CO_2$ being dispersed in the atmosphere by wind dynamics.

**3 Minor Comments**

- The authors do make it clear that they plan to update the enhancements data set when new input products are made available from either NOAA or for OCO. The newest version of NASA ACOS OCO-3 $XCO_2$ is v11, which replaced v10.4 that was used in the current study. Improvements include, among other things, changes to the OCO-3 geolocation. So the current $\Delta XCO_2$ product is already out-of-date, although I would expect changes to be modest based on OCO-3 v11.

- L90-91: I think this is the first time where it is mentioned that the global $\Delta XCO_2$ is provided only over land.

**4 Technical Comments**

- L330-331: Error in author list for this citation. Remove the "S., Washington, W.-C., and Baltimore, D. C" as those are place-names, not authors!

- L349: The current link to the article does not work. Try using just [https://doi.org/10.1073/PNAS.1702393115](https://doi.org/10.1073/PNAS.1702393115).

- L389: the current link does not work. Should be https://doi.org/10.1016/S0921-8181(00)00050-3.

**5 Citations**

- Byrne, National $CO_2$ budgets (2015–2020) inferred from atmospheric $CO_2$ observations in support of the global stocktake, Earth System Science Data, 2023, https://doi.org/10.5194/essd-15-963-2023

- S. Houweling, An intercomparison of inverse models for estimating sources and sinks of $CO_2$ using GOSAT measurements, J. Geophysical Research - Atmospheres, 2015, https://doi.org/10.1002/2014JD022962.

- Laughner, A new algorithm to generate a priori trace gas profiles for the GGG2020 retrieval algorithm, Atmos. Meas. Techniques, 2023, https://amt.copernicus.org/articles/16/1121/2023/.

- Laughner, The Total Carbon Column Observing Network's GGG2020 data version, Atmos. Meas. Techniques, 2024, https://essd.copernicus.org/articles/16/2197/2024/.

- Lindqvist, Does GOSAT capture the true seasonal cycle of carbon dioxide?, Atmos. Chem. Phys., 2015, https://www.atmos-chem-phys.net/15/13023/2015/

- Nguyen, Intercomparison of Remote Sensing Retrievals: An Examination of Prior-Induced Biases in Averaging Kernel Corrections, Remote Sensing, 2020, https://doi.org/10.3390/rs12193239

- O'Dell, Improved retrievals of carbon dioxide from Orbiting Carbon Observatory-2 with the version 8 ACOS algorithm, Atmos. Meas. Techniques, 2018, https://doi.org/10.5194/amt-11-6539-2018

- Pandy, Toward Low-Latency Estimation of Atmospheric $CO_2$ Growth Rates Using Satellite Observations: Evaluating Sampling Errors of Satellite and In Situ Observing Approaches, AGU Advances, 2024, https://doi.org/10.1029/2023AV001145

- Rastogi, B., Evaluating consistency between total column $CO_2$ retrievals from OCO-2 and the in-situ network over North America: Implications for carbon flux estimation. Atmospheric Chemistry and Physics, 2021, https://doi.org/10.5194/acp-21-14385-2021.

- Roten, Constraining Sector-Specific $CO_2$ Fluxes Using Space-Based XCO2 Observations Over the Los Angeles Basin, Geophysical Research Letters, 2023, https://agupubs.onlinelibrary.wiley.com/doi/abs/10.1029/2023GL104376

- Taylor, Evaluating the consistency between OCO-2 and OCO-3 $XCO_2$ estimates derived from the NASA ACOS version 10 retrieval algorithm, Atmos. Meas. Techniques, 2023, https://doi.org/10.5194/amt-16-3173-2023

- Wu, Towards sector-based attribution using intra-city variations in satellite-based emission ratios between $CO_2$ and CO, Atmospheric Chemistry and Physics, 2022, https://acp.copernicus.org/articles/22/14547/2022/

- Ye, Constraining Fossil Fuel CO2 Emissions From Urban Area Using OCO-2 Observations of Total Column CO2, Journal of Geophysical Research: Atmospheres, 2020, https://agupubs.onlinelibrary.wiley.com/doi/abs/10.1029/2019JD030528